# Serum 25-hydroxyvitamin D Concentration Significantly Decreases in Patients with COVID-19 Pneumonia during the First 48 Hours after Hospital Admission

**DOI:** 10.3390/nu14122362

**Published:** 2022-06-07

**Authors:** Juraj Smaha, Martin Kužma, Peter Jackuliak, Samuel Nachtmann, Filip Max, Elena Tibenská, Neil Binkley, Juraj Payer

**Affiliations:** 15th Department of Internal Medicine, Comenius University Faculty of Medicine, University Hospital, Ruzinovska 6, 826 06 Bratislava, Slovakia; martin.kuzma@fmed.uniba.sk (M.K.); peter.jackuliak@fmed.uniba.sk (P.J.); samuel.nachtmann@gmail.com (S.N.); elena.tibenska@medirex.sk (E.T.); payer@ru.unb.sk (J.P.); 2Department of Pharmacology and Toxicology, Comenius University Faculty of Pharmacy, Odbojarov 10, 832 32 Bratislava, Slovakia; max1@uniba.sk; 3Medirex, a.s., Galvaniho 17/C, 820 16 Bratislava, Slovakia; 4Department of Medicine, Geriatrics Faculty, Medical Sciences Center, University of Wisconsin, 1300 University Ave, Madison, WI 53706-1510, USA; nbinkley@wisc.edu

**Keywords:** vitamin D, 25-hydroxyvitamin D, COVID-19, inflammation, pneumonia, SARS-CoV-2

## Abstract

It is unclear how ongoing inflammation in Coronavirus Disease 2019 (COVID-19) affects 25-hydroxyvitamin D (25[OH]D) concentration. The objective of our study was to examine serum 25(OH)D levels during COVID-19 pneumonia. Patients were admitted between 1 November and 31 December 2021. Blood samples were taken on admission (day 0) and every 24 h for the subsequent four days (day 1–4). On admission, 59% of patients were 25(OH)D sufficient (>30 ng/mL), and 41% had 25(OH)D inadequacy (<30 ng/mL). A significant fall in mean 25(OH)D concentration from admission to day 2 (first 48 h) was observed (30.7 ng/mL vs. 26.4 ng/mL; *p* < 0.0001). No subsequent significant change in 25(OH)D concentration was observed between day 2 and 3 (26.4 ng/mL vs. 25.9 ng/mL; *p* = 0.230) and day 3 and day 4 (25.8 ng/mL vs. 25.9 ng/mL; *p* = 0.703). The absolute 25(OH)D change between hospital admission and day 4 was 16% (4.8 ng/mL; *p* < 0.0001). On day 4, the number of patients with 25(OH)D inadequacy increased by 18% (*p* = 0.018). Therefore, serum 25(OH)D concentration after hospital admission in acutely ill COVID-19 patients should be interpreted with caution. Whether low 25(OH)D in COVID-19 reflects tissue level vitamin D deficiency or represents only a laboratory phenomenon remains to be elucidated in further prospective trials of vitamin D supplementation.

## 1. Introduction

The role of vitamin D metabolites as a potentially modifiable risk factor in Coronavirus Disease 19 (COVID-19) infection was suggested early in the pandemic [1]. Since then, many studies relating vitamin D status with COVID-19 severity and outcome have been published. Regarding COVID-19, low 25-hydroxyvitamin D (25[OH]D) levels have been associated with a higher risk of testing positive [2], the severity of infection [3], higher need for invasive ventilation [4], and higher mortality [5]. Our previous study found that serum 25(OH)D concentration measured at admission in patients with severe COVID-19 is an independent risk factor for mortality. Moreover, patients with severe vitamin D deficiency, (i.e., 25[OH]D < 12 ng/mL) had a higher viral load, higher Charlson comorbidity index, and an 11% increase in mortality rate than patients with a serum 25(OH)D concentration above 12 ng/mL [5].

COVID-19 infection causes a systemic hyperimmune response triggered by the hypoxic environment due to respiratory failure and the ensuing cytokine storm [6]. During this acute phase, metabolic changes in macronutrients, as well as micronutrients, could be expected; vitamin D metabolites are no exception to this [7]. Serum 25(OH)D concentration reflects vitamin D body stores, and, under normal circumstances, it is the best indicator of vitamin D status [8]. However, the role of 25(OH)D concentration as a biomarker for acute inflammatory illness is controversial. The literature suggests that significant variations in 25(OH)D levels may occur within hours in acutely ill patients. Many pathophysiological mechanisms have been proposed, e.g., a direct effect of inflammation, hemodilution, decreased synthesis of binding proteins, or renal wasting of 25(OH)D [9]. As such, a single-point assessment of 25(OH)D may be inaccurate in estimating vitamin D status during acute phase response [7]. Consistent with this, in one study, the prevalence of vitamin D inadequacy increased from 38% before total hip arthroplasty even to 68% the day after surgery [10].

To the best of our knowledge, there is no study prospectively evaluating serum 25(OH)D levels during acute COVID-19 infection. The objective of our study was to examine whether serum 25(OH)D levels change during the evolution of acute COVID-19 pneumonia and to explore the possibility of determining an optimal time window for 25(OH)D concentration assessment during acute illness.

## 2. Materials and Methods

This study was undertaken as a prospective cohort study. Patients with acute COVID-19 pneumonia hospitalized in the internal medicine department between 1 November 2021 and 31 December 2021 were recruited. Exclusion criteria were as follows:Patients with no need for supplemental oxygen;Patients not meeting the criteria for severe disease;COVID-19 pneumonia was not the primary diagnosis upon admission;Patients with another acute infection, (e.g., urinary tract infection) during the monitoring period.

Demographic characteristics, comorbidities, hematological and biochemical laboratory results on admission, information regarding the intensity of care during hospitalization, and pharmacological treatment before (including vitamin D supplementation) and during hospitalization were collected from electronic medical records and discharge summaries by two physicians using a standardized approach. All patients received pharmacological and supportive measures according to interim COVID-19 guidance of treatment approved by the University Hospital Bratislava. These guidelines were based on current Centers for Disease Control and Prevention (USA) recommendations for treatment. (https://www.cdc.gov/coronavirus/2019-ncov/hcp/clinical-guidance-management-patients.html, accessed on 1 October 2021). Patients were not supplemented with vitamin D preparations during the monitoring period. Severe infection was defined as clinical signs of pneumonia and one of the following: respiratory rate >30 breaths/minute; severe respiratory distress; or oxygen saturation <90% on room air.

A first venous blood sample was taken on admission (day 0) and then every 24 h for the subsequent four days: the second sample at the 24th hour (day 1), the third sample at the 48th hour (day 2), the fourth sample at the 72nd hour (day 3), and the fifth sample at the 96th hour (day 4), respectively.

Serum 25(OH)D concentrations (in ng/mL) were obtained using an automated electrochemiluminescence system (Eclesys Vitamin D Total II, 2019, Roche Diagnostics GmBH, Mannheim, Germany). The detection limit of serum 25(OH)D was 3 ng/mL. The complete blood count testing was performed on an automated analyzer (Mindray BC-6800 Plus Auto Hematology Analyzer). CRP was analyzed by immunoturbidimetric quantitative assay (Roche, CRP4 Cobas, module c 501). Other tests such as liver tests, kidney functions, and serum minerals were performed on each patient and assessed with commercial standardized tests. The presence of the SARS-COV-2 virus was assessed by a real-time reverse transcriptase-polymerase chain reaction (RT-PCR) test on nasopharyngeal swab. Blood oxygen saturation was assessed using arterial blood sampling ±2 h from venous blood sampling.

Statistical analysis was performed using the statistics software Analyse-it (Leeds, UK) v 5.40.2 or R (v 3.6.0). Continuous data were expressed as the mean ± standard error of the mean (SEM) if normally distributed or as median and interquartile range if not normally distributed. The Shapiro–Francia test was used to test the normality of the distributions of studied parameters. Changes between time points were assessed with Student’s paired *t*-test and Wilcoxon test in normally and not normally distributed variables, respectively. Relationships between investigated parameters were calculated using the Pearson correlation coefficient or binominal logistic regression. *p*-values less than 0.05 were considered to be statistically significant.

The study was conducted in accordance with the Declaration of Helsinki and approved by the Institutional Ethics Committee of University Hospital Bratislava (ethical approval code: EK/011/2021) on 1 January 2021.

## 3. Results

The baseline characteristics of the patients are shown in Table 1. Twenty-two patients were included (six females, and sixteen males; the median age of 60.6 years). Twelve patients (55%) have a history of arterial hypertension, seven patients (32%) have a history of diabetes mellitus, two patients (9%) have a history of coronary artery disease, and five patients (23%) have a history of chronic kidney disease. The mean value of body mass index (BMI) was 29.74 kg/m^2^; ten patients (45%) were overweight (BMI > 25 kg/m^2^) and eight patients were obese (BMI > 30 kg/m^2^). All patients required supplemental oxygen, of whom twelve patients (55%) needed high-flow oxygen via nasal cannula. Two patients required invasive mechanical ventilation. Four patients eventually died after completing the protocol.

The patients were recruited between 1 November and 31 December. Symptoms of COVID-19 (fever, chills, cough, nausea, malaise, myalgias, cefalea) were present 7.45 days before hospitalization (mean value). Dyspnea had been present for more than 48 h in 36% and less than 48 h in 64% of the patients. The baseline 25(OH)D concentration was 30.71 ng/mL; 59% of patients were 25(OH)D sufficient (>30 ng/mL), 18% were 25(OH)D insufficient (30–20 ng/mL) and 5% were 25(OH)D deficient (<20 ng/mL). A total of 45% of patients used vitamin D supplements before hospitalization. There was no association between 25(OH)D status upon admission and mortality, the need for high flow nasal oxygen, or duration of symptoms. The differences in baseline 25(OH)D concentration between selected subgroups of patients upon admission are displayed in Table 2.

A significant fall in mean 25(OH)D concentration from admission to day 2 (first 48 h) was observed (30.7 ng/mL vs. 26.4 ng/mL; *p* < 0.0001). No subsequent significant fall in 25(OH)D concentration was observed between day 2 and 3 (26.4 ng/mL vs. 25.9 ng/mL; *p* = 0.2300) and day 3 and day 4 (25.8 ng/mL vs. 25.9 ng/mL; *p* = 0.7026). The evolution of serum 25(OH)D levels change is displayed in Figure 1.

Upon admission, serum 25(OH)D levels were higher in males compared to females, although this difference was not statistically significant (*p* = 0.43). A decline in serum 25(OH)D levels was observed in males, as well as females. In males, the decline in 25(OH)D concentration in the first 48 h was more significant compared to females (*p* < 0.005 vs. *p* < 0.05, respectively). The decline between days 2 and 3 was observed in males, but no further decline was seen in females. The 25(OH)D kinetics in males and females is displayed in Figure 2, respectively.

In the majority of patients (n = 17), serum 25(OH)D levels decreased during the monitoring period, see Figure 3.

The absolute 25(OH)D change between hospital admission and day 4 was 4.8 ng/mL (*p* < 0.0001) and was not associated with mortality (*p* = 0.211), the need for high flow oxygen (*p* = 0.647), or duration of symptoms (*p* = 0.14). On day 4, the number of patients with 25(OH)D inadequacy (<30 ng/mL) increased by 18% (*p* = 0.018). The kinetics of the 25(OH)D concentration during hospitalization was compared between survivors and non-survivors (Figure 4), between patients treated with high flow oxygen and conventional oxygen (Figure 5), and between shorter and longer duration of symptoms than the median (Figure 6). In survivors, the decline in 25(OH)D levels during the first 48 h was highly statistically significant (*p* < 0.0001) in comparison to non-survivors, where no significant change in 25(OH)D concentration was observed (*p* = 0.2). Irrespective of the mode of oxygen therapy, a statistically significant decline in 25(OH)D levels during the first 48 h of hospitalization was observed (high-flow oxygen: *p* = 0.018; conventional oxygen: *p* = 0.04). Patients with a shorter duration of symptoms before hospitalization have a slightly higher decline in 25(OH)D concentration during the first 48 h compared to patients with a longer duration of symptoms (<7 days of symptoms: *p* = 0.001 vs. >7 days of symptoms: *p* = 0.01).

A significant fall in creatinine, albumin, hemoglobin, and hematocrit was observed (day 1 vs. day 5, all *p* < 0.05). Both albumin and hemoglobin significantly decreased during the first 48 h after the hospital admission (*p* < 0.05 and *p* < 0.005, respectively). For albumin and hemoglobin concentration kinetics during the study period, see Appendix A.

Regarding the markers of inflammation, C-reactive protein (CRP) and interleukin-6 (IL-6) significantly decreased during the monitoring period (day 0 vs. day 4, both *p* < 0.0005). We observed a highly significant rapid fall in IL-6 concentration in the first 24 h, followed by an increase in concentration and a slight subsequent decrease. For CRP and IL-6 concentration kinetics during the study period, see Appendix A. There was a significant increase in neutrophils between admission and day 1 and a subsequent decrease in the number of neutrophils from day 1 to day 3 (all *p* < 0.0001). Lymphocytes did not change significantly during the first 48 h; after the first 48 h, a statistically significant increase in lymphocyte concentration was observed (day 0 vs. day 4; *p* < 0.05). Monocytes increased significantly during the monitoring period (*p* < 0.005). The kinetics of the concentration of neutrophils, lymphocytes, and monocytes are displayed in Appendix A. The changes in concentration of all parameters during the monitoring period are displayed in Table 3. For all laboratory parameters of each patient during the monitoring period see Appendix A.

## 4. Discussion

In this study, we found a significant reduction in circulating 25(OH)D concentration of ~16% within two days of hospitalization in patients with severe COVID-19 pneumonia. As a result, the proportion of patients with vitamin D inadequacy increased from 41% to 59%. In this pilot study, the change in 25(OH)D was unrelated to changes in CRP and IL-6. Serum 25(OH)D levels tend to be significantly higher in males than in females across all BMI groups [11]. In our cohort of patients, we also found a difference in 25(OH)D concentration between females and males upon admission. Females had lower 25(OH)D levels, although this difference was not statistically significant. A significant decline in 25(OH)D concentration during the first 48 h after the hospital admission was observed in both sexes. In males, this decline was more significant and observed even after the first 48 h.

The existing literature associates inflammation of various causes with low 25(OH)D. Most studies to date considered healthcare-associated intervention, i.e., elective surgery, as an inflammatory stimulus. Except for one study, which retrospectively assessed 25(OH)D concentration during malarial infection [12], there is no study investigating 25(OH)D changes in the context of acute inflammatory reaction caused by an infectious disease.

Serum 25(OH)D concentration decreases hours (6–48) after the surgical procedure, and the maximum change in 25(OH)D concentration could be as high as 40% compared to the baseline status [13,14]. CRP was most commonly used as an inflammatory marker. Due to the correlation of increased CRP concentration with reduced 25(OH)D levels in these studies, many have suggested that serum 25(OH)D is simply a negative acute phase reactant [14,15]. However, the observed changes could result from surgery and anesthetic management rather than inflammation itself [10]. Nevertheless, a decrease in 25(OH)D levels, accompanied by an increase in CRP, was also observed during the first days after acute pancreatitis [16] and after the intravenous infusion of bisphosphonates [17].

In contrast, neither change in 25(OH)D serum levels nor significant correlation between markers of inflammation were observed in patients after acute myocardial infarction [18] and severe malarial infection [12], respectively. However, in both studies, serum 25(OH)D levels were measured several days after symptom onset, which might blunt the ability to see a decline.

Interestingly, even though our patients were symptomatic for 7.45 days and experienced dyspnea for 2.45 days before hospitalization, we still observed a significant decline in 25(OH)D. The CRP was at its peak at admission, and its concentration decreased during the study period, presumably because of treatment with immunomodulatory drugs. Notably, severe illness in people with COVID-19 typically occurs approximately 8–12 days after symptoms onset [19]. The most common symptom is dyspnea, accompanied by hypoxemia. Importantly, in the severe phase of COVID-19, the pulmonary lesions generally peaked 6–11 days after the symptom onset [20]. It is thus possible that excessive local pulmonary inflammation was close to its peak at the time of hospital admission, and functional vitamin D deficiency was observed.

In this regard, it is plausible that vitamin D deficiency would be related to tissue requirement. In such a case, the circulating 25(OH)D pool represents a substrate reservoir for conversion to active metabolites (1,25[OH]2D) at the pulmonary tissue level during times of acute stress [21]. Several previous studies have suggested that the vitamin D metabolic pathways may influence the development of acute respiratory distress syndrome (ARDS) by various mechanisms [22,23]. Interestingly, Abrishami et al. have observed that higher levels of 25(OH)D were associated with significantly less total lung involvement on chest computed tomography in hospitalized COVID-19 patients [24].

Factors other than inflammation were also identified as a potential cause of the 25(OH)D decline during acute illness. For example, intravenous fluid administration has been associated with reduced 25(OH)D levels. Krishnan et al. showed that acute fluid loading rather than inflammation might have a more profound effect on 25(OH)D concentration during the early phase of acute illness [25]. However, Reid et al. observed that after the administration of 3 liters of intravenous fluid over 24 h after knee arthroplasty, the concentration of 25(OH)D dropped by 40% and was accompanied only by a 15% reduction in albumin concentration and 10% in vitamin D binding protein concentration (VDBP) [14]. As such, acute fluid loading does not seem to play a significant role in our cohort of patients with considerably less intravenous fluid given during the first 48 h after hospital admission.

Hypoalbuminemia is a common condition in patients with serious illnesses [26]. Albumin binds approximately 15% of serum 25(OH)D; thus, hypoalbuminemia of critically ill patients could be responsible for approximately 15% of the variation 25(OH)D [27]. Binding proteins are also essential for 25(OH)D reabsorption at the renal tubules; therefore, loss of these proteins may lead to subsequent renal wasting of 25(OH)D during acute illness [7]. Indeed, the decrease in serum VDBP during systemic inflammatory response is significantly associated with increased urinary loss of VDBP [13].

It has been suggested that glucocorticoid administration may decrease 25(OH)D levels [28]. All of our patients received six milligrams of intravenous dexamethasone daily during the study duration. Dexamethasone increases renal expression of vitamin D-24-hydroxylase, which degrades vitamin D metabolites such as 25(OH)D [29]. After 24 h of therapy with dexamethasone, a significant abundance of 24-hydroxylase was observed [30]. Additionally, glucocorticoids enhance direct 24-hydroxylase transcription via cooperation between glucocorticoid receptor C/EBPbeta and vitamin D receptor (VDR) [31]. Thus, the possibility of lowering 25(OH)D levels because of the infusion of glucocorticoids cannot be ruled out in patients with COVID-19 and hypoxemia. Consistent with this, an observed increase in the number of neutrophils could also be connected with systemic glucocorticoid administration. The white blood count rises after glucocorticoid administration mainly because of neutrophilic migration from the endothelial lining of the blood vessels, which manifests within 5–24 h following administration and can persist during therapy [32]. However, specifically in critically ill COVID-19 patients, treatment with dexamethasone was actually associated with lower neutrophil proportions in bronchoalveolar lavage compared to patients without dexamethasone, thus highlighting the critical role of neutrophils in the pathophysiology of ARDS in COVID-19 [33].

Our study has several limitations. Above all, we evaluated a small group of patients. Our primary aim was to observe changes in serum 25(OH)D levels; thus, this present study was underpowered to see whether an association between mortality and magnitude of change in 25(OH)D serum levels exists. Secondly, we did not have a control group of patients. Additionally, treatment with glucocorticoids is currently a therapeutic option supported by the highest level of evidence; therefore, we also did not have a cohort of patients without i.v. glucocorticoid treatment. Thus, we cannot rule out the possible effect of hospitalization or i.v. glucocorticoid administration on the evaluated laboratory parameters. Thirdly, we did not know the serum 25(OH)D concentration of patients prior to hospitalization. We also did not evaluate VDBP; thus, we cannot exclude some variations of 25(OH)D connected with a possible decrease in VDBP. Our study also has several strengths. We have prospectively evaluated a homogeneous, clearly defined group of patients with acute severe COVID-19 pneumonia. The 25(OH)D concentration was evaluated with the same method and technique in all patients at precisely defined periods. To the best of our knowledge, this is the first study in which a change of 25(OH)D concentration was evaluated for a longer period of time in hospitalized patients with COVID-19. Additionally, this is also the first study with a prospective design regarding changes of 25(OH)D in the context of an acute infectious disease.

In conclusion, this study showed that serum 25(OH)D concentration decreases significantly during the first 48 h after hospital admission in acutely ill COVID-19 patients. The number of patients with 25(OH)D inadequacy increased by 18% during the monitoring period. After the first 48 h, no significant change in 25(OH)D concentration was observed, which could have practical implications for vitamin D status assessment during acute COVID-19. Whether low 25(OH)D in COVID-19 reflects functional vitamin D deficiency and has a causal link to worse prognosis in COVID-19 or represents only a laboratory phenomenon remains to be elucidated in further prospective randomized trials of vitamin D supplementation.

## Figures and Tables

**Figure 1 nutrients-14-02362-f001:**
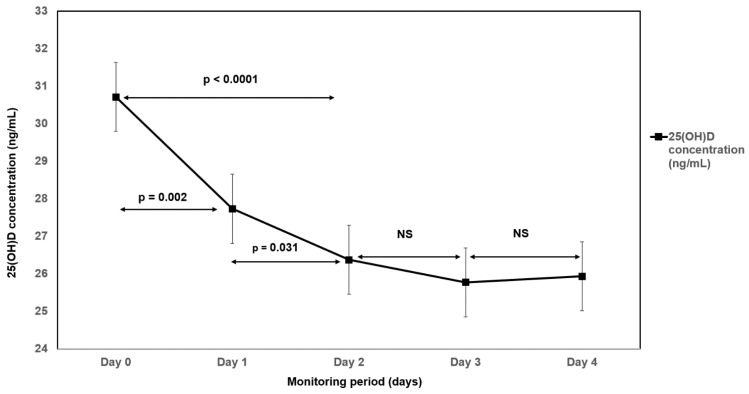
The evolution of serum 25-hydroxyvitamin D (25[OH]D) levels change during the study period.

**Figure 2 nutrients-14-02362-f002:**
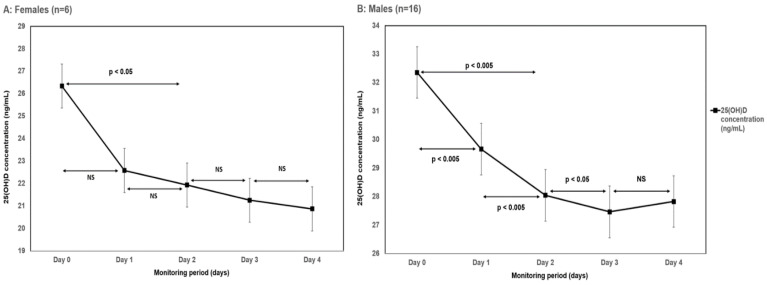
Serum 25-hydroxyvitamin D (25[OH]D) levels kinetics during the study period in females (**A**) and males (**B**), respectively.

**Figure 3 nutrients-14-02362-f003:**
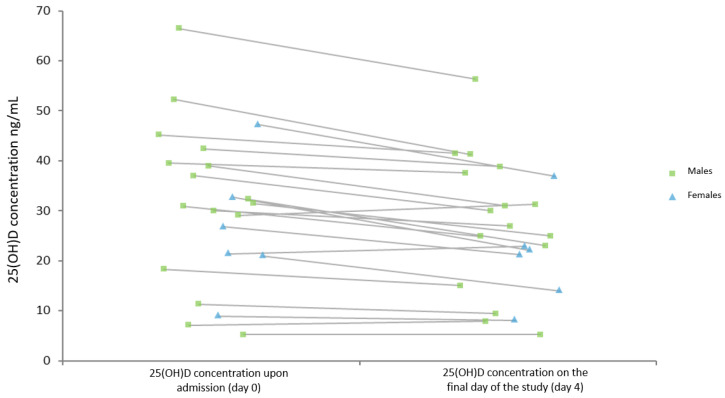
The comparison of 25-hydroxyvitamin D (25[OH]D) status in all patients between day of admission and day 4 of the study.

**Figure 4 nutrients-14-02362-f004:**
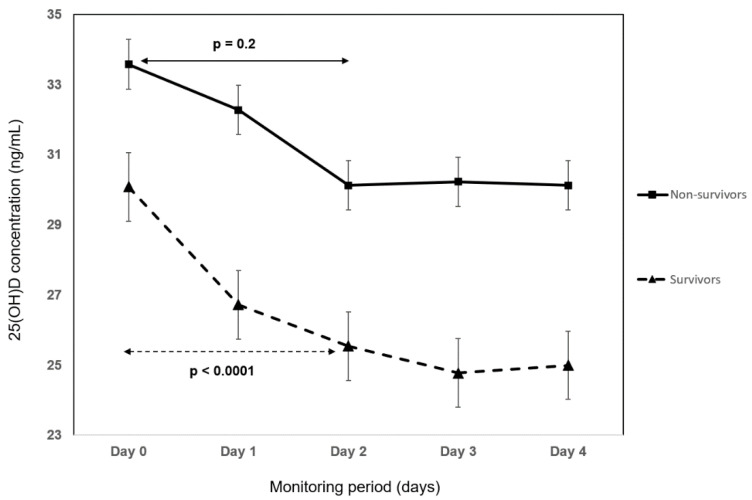
The kinetics of 25-hydroxyvitamin D (25[OH]D) concentration during hospitalization between survivors and non-survivors. *p*-values of the changes during the first 48 h are displayed.

**Figure 5 nutrients-14-02362-f005:**
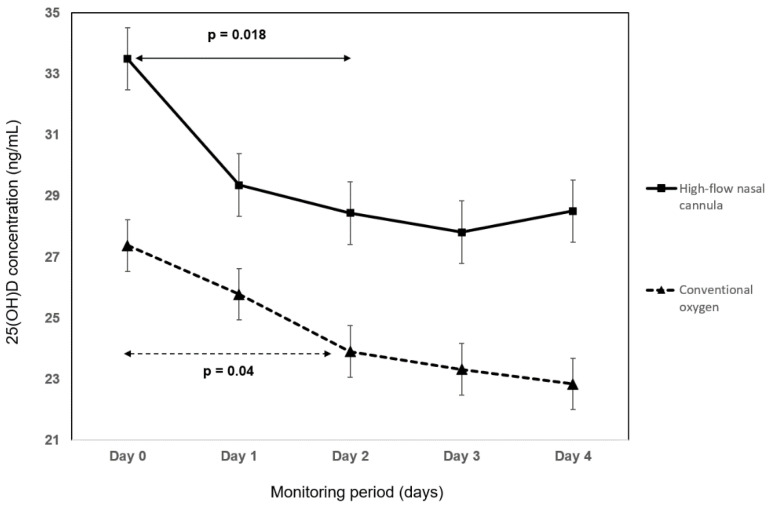
The kinetics of 25-hydroxyvitamin D (25[OH]D) concentration between patients treated with high-flow oxygen and conventional oxygen. *p*-values of the changes during the first 48 h are displayed.

**Figure 6 nutrients-14-02362-f006:**
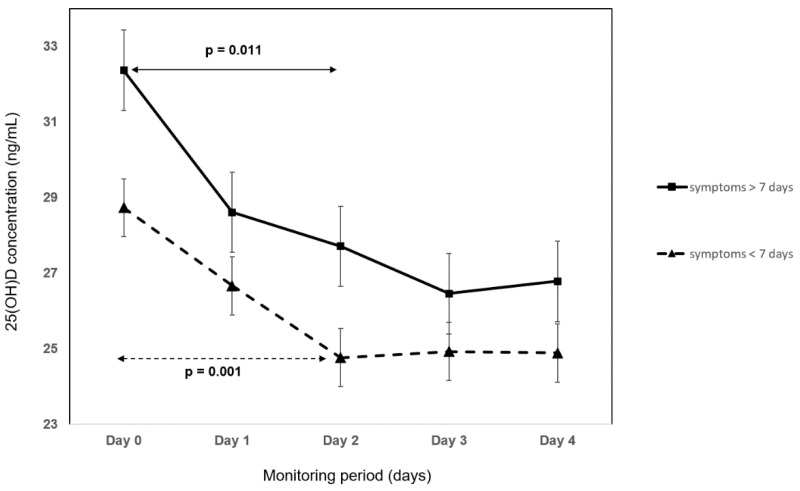
The kinetics of 25-hydroxyvitamin D (25[OH]D) concentration during hospitalization between shorter and longer duration of symptoms than median (median = 7 days).

**Table 1 nutrients-14-02362-t001:** Baseline characteristics of participants.

Variable	n = 22	Females (n = 6)	Males (n = 16)
Females/males (n, %)	6 (27%)/16 (73%)	N/A	N/A
Survivors/non-survivors (n, %)	18 (82%)/4 (18%)	5 (83%)/1 (17%)	13 (81%)/3 (19%)
Age (years) (mean ± SD)	60.5 ± 14.4	66.8 ± 12	58.3 ± 15
Body mass index (kg/m^2^), median (IQR)	29.74 (7.3)	26.5 (8.8)	28.9 (7.34)
Arterial hypertension (n, %)	12 (55%)	3 (50%)	9 (56%)
Diabetes mellitus (n, %)	7 (32%)	2 (33%)	5 (31%)
Coronary artery disease (n, %)	2 (9%)	1 (17%)	1 (6%)
Chronic kidney disease (n, %)	5 (23%)	2 (33%)	3 (19%)
Number of symptomatic days before hospitalization, mean ± SD	7.45 ± 3.1	7.6 ± 3.4	7 ± 2.6
Days of dyspnea before hospitalization, median (IQR)	2.45 (1.1)	2 (1.0)	2 (3.2)
Need for supplemental oxygen (n, %)	22 (100%)	6 (100%)	16 (100%)
Need for high flow oxygen (n, %)	12 (55%)	3 (33%)	9 (56%)
Number of days of high flow oxygen, n (IQR)	7 (6)	4 (6)	2.5 (6.1)
Invasive mechanical ventilation (n, %)	2 (9%)	0 (0%)	2 (13%)
Vitamin D supplementation before hospitalization (n, %)	10 (45%)	2 (33%)	8 (50%)

**Table 2 nutrients-14-02362-t002:** The difference in 25-hydroxyvitamin D (25[OH]D) concentration between selected subgroups of patients upon admission.

Variable	25(OH)D (ng/mL) upon Admission	*p*-Value
Sex		0.43
Males	32.3 ± 16.2	
Females	26.3 ± 12.9	
Body mass index		
>30 kg/m^2^	27.5 ± 16.2	0.47
<30 kg/m^2^	32.5 ± 15.1	
End of hospitalization		0.69
Death	33.6 ± 13.2	
Discharge	30.1 ± 16.1	
Arterial hypertension		0.57
Yes	28.9 ± 10.4	
No	32.7 ± 20.2	
Diabetes mellitus		0.73
Yes	29.03 ± 12.5	
No	31.5 ± 16.9	
Coronary artery disease		0.61
Yes	25.2 ± 5.4	
No	31.2 ± 16.1	
Chronic kidney disease		0.18
Yes	22.5 ± 10.2	
No	33.1 ± 16.1	
Chronic pulmonary disease		0.26
Yes	18.9 ± 19.4	
No	31.8 ± 15.01	
Need for high flow oxygen		0.36
Yes	33.5 ± 16.6	
No	27.4 ± 13.8	
Vitamin D supplementation before hospitalization		0.0005
Yes	41.85 ± 11.45	
No	21.4 ± 11.7	

**Table 3 nutrients-14-02362-t003:** The changes in concentration of all parameters during the monitoring period compared to the baseline value on day 0 (admission).

Variable (Mean)	Day 0	Day 1	Day 2	Day 3	Day 4
25(OH)D (ng/mL) ± SD	30.7 ± 15.4	27.7 ± 13.5 **	26.4 ± 12.7 **	25.8 ± 12.5 **	25.9 ± 13.05 **
Neutrophils (10 × 9/L) (IQR)	6.0 (6.7)	8.2 (8.1) **	9.4 (6.3) *	8.7 (5.2)	7.9 (4.6)
Lymphocytes (10 × 9/L)	0.90 ± 0.42	0.96 ± 0.4	0.99 ± 0.53	1.19 ± 0.62 *	1.29 ± 0.68 *
Monocytes (10 × 9/L) (IQR)	0.32 (0.23)	0.54 (0.3) **	0.5 (0.3) **	0.58 (0.27) **	0.53 (0.27) **
C-reactive protein (mg/L) (IQR)	162.8 (102.3)	94.5 (100.5) **	66.2 (72.2) **	51.2 (53.3) **	28.9 (43.1) **
Interleukin-6 (ng/L) (IQR)	69.8 (148)	30.3 (43.4) **	35.2 (69.7) *	31.6 (61.7)*	11.1 (44.8) **
Procalcitonin (ng/mL) (IQR)	0.28 (0.8)	0.19 (0.34) *	0.13 (0.23)	0.12 (0.26)*	0.095 (0.21) **
Lactate dehydrogenase (IU) ± SD	8.85 ± 4.4	8.83 ± 4.5	8.28 ± 4.1	8.28 ± 4.15	7.45 ± 3.82 *
Neutrophil to lymphocyte ratio (IQR)	7(10.2)	10.5(8.1) *	10(11)	7.5(6.5)	6.5 (10)
Calcium (mmol/L) (IQR)	2.12 (0.26)	2.16 (0.18)	2.02 (0.19)	2.17 (0.15)	2.16 (0.17)
Ionized calcium (mmol/L) ± SD	1.137 ± 0.1	1.16 ± 0.07 *	1.17 ± 0.1 *	1.18 ± 0.08 *	1.18 ± 0.1 *
Blood urea nitrogen (mmol/L) (IQR)	6.6 (6.5)	7.5 (6.8)	8.5 (8.2)	8.2 (3.8)	7.6 (3.1)
Creatinine (umol/L) (IQR)	94.3 (43.3)	73.1 (31.1)	81.2 (19.6)	78.2 (11.6)	79.3 (15.5) *
Albumin (g/L) ± SD	34.7 ± 4.6	32.9 ± 3.8 *	32.1 ± 3.8 *	31.9 ± 3.9 *	31.8 ± 4.1 *
Hemoglobin (g/L) ± SD	138 ± 15	136 ± 15	132 ± 16.8 **	133 ± 15.9 *	133 ± 17 *
Hematocrit ± SD	0.42 ± 0.05	0.41 ± 0.05	0.403 ± 0.05 **	0.409 ± 0.05 *	0.409 ± 0.05 *
Alanine aminotransferase (ukat/L)	1.43 ± 2.03	1.5 ± 1.7	1.65 ± 2.04	1.91 ± 1.83	2.1 ± 2.17
Alkaline phosphatase (ukat/L) (IQR)	1.5 (1.38)	1.44 (1.02)	1.69 (0.92)	1.71 (0.96)	1.61 (0.91)

** *p* < 0.005. * *p*< 0.05.

## Data Availability

The data presented in this study are available on request from the corresponding author.

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
