# Peer review of "Serum 25-hydroxyvitamin D Concentration Significantly Decreases in Patients with COVID-19 Pneumonia during the First 48 Hours after Hospital Admission"

_nutrients, 2022, doi:10.3390/nu14122362_

Round 1

Reviewer 1 Report

Review of “Serum 25-hydroxyvitamin D Concentration Significantly Decreases in Patients with COVID-19 Pneumonia During the First 48 Hours after Hospital Admission” by Juraj Smaha et al.

This is an interesting study of serum 25(OH)D concentration of COVID-19 patients after hospital admission. It provided data of 22 patients and claims that serum 25(OH)D levels were reduced in these patients specially during the initial two days. Since the effects of 25(OH)D levels to COVID-19 infection risk, hospitalization, etc. is a topic under focus, this study could be interested by many researchers in the same field. Evidence from this article could be a useful reference in the treatment of COVID19. Here I listed few questions that need to be addressed before publishing:

1. A control group was missing from the data. The reduction of serum 25-hydroxyvitamin D concentration may be due to the hospital admission. Do patients without COVID-19 have similar serum 25-hydroxyvitamin D reduction after hospital admission? This need to be assessed before making a conclusion.

2. The patient group has 6 females and 16 males. A study by Giovanna Muscogiuri et al. 2019 showed that vitamin D levels are significantly higher in males than in females across all BMI groups. Thus, female and male patients in this study need to be investigated separately.

3. Since only a small group of 22 patients are evaluated, the authors can follow serum 25(OH)D levels of each patient. How many of these patients showed a reduction of 25(OH)D compare with their status on day 0? A table or supplement table of all parameters of each patient during the monitoring period would also provide more information.

Reviewer 2 Report

Overall: The authors found a significant reduction in circulating 25(OH)D concentration of ~16% within two days of hospitalization in patients with severe COVID-19 pneumonia, which was unrelated to changes in C-reactive protein and IL-6. This study is very interesting, although data were obtained from only twenty-two patients with severe COVID-19 pneumonia.

Major points

1.     Statistics

Which statistics were used to calculate following p-values?

Line 141 ~ 143

The absolute 25(OH)D change between hospital admission and day 4 was 4.8 ng/mL and was not associated with mortality (p=0.2113), need for high flow oxygen (p=0.6467), or duration of symptoms (p=0.14).

The kinetics of the 25(OH)D value during admission should be compared between survived and died patients, between needed and not-needed high-flow oxygen via nasal cannula, and between shorter and longer duration of symptoms than median, by using repeated measure analysis of variance (ANOVA). Even not significant, it is better to be shown by graphics like Figure 1.

2.     Tables

The Shapiro-Francia test was used to test the normality of the distributions of studied parameters. Parametric and nonparametric continuous variables with normal and nonnormal distributions should be expressed by mean the mean ± standard deviation (or error) and by median with interquartile range, respectively.

Minor points

1.      Figures

The kinetics of 25(OH)D were duplicated in Figure 1 and Figure 2 as well as in Table 3. Following figures are suggested;

Figure 1. Changes of serum 25(OH)D levels during the study period.

Figure 2. Changes of neutrophils, lymphocytes, and monocytes counts during the study period.

Figure 3. Changes of serum CRP and interleukin-6 levels during the study period.

Figure 4. Changes of serum albumin and hemoglobin levels during the study period.

2.      Vitamin D supplementation

Could you analyze the effects of vitamin D supplementation before admission on clinical outcomes such as mortality, need for high flow oxygen, or duration of symptoms?

3.      Serum 25(OH)D level on admission

Could you analyze the associations of serum 25(OH)D level on admission with clinical outcomes such as mortality, need for high flow oxygen, or duration of symptoms?

4.      Study design, e.g., retrospective cohort study, as well as Ethics Statement should be shown.

Round 2

Reviewer 1 Report

The authors have addressed all my questions and comments, and the paper is improved, thus I would recommend it for publish.

Reviewer 2 Report

Authors responded to the reviewers' questions.